# Bronchoalveolar Lavage Cytology in Severe Equine Asthma: Cytocentrifugated versus Sediment Smear Preparations [note 1]

**DOI:** 10.3390/vetsci10080527

**Published:** 2023-08-16

**Authors:** Maria Morini, Francesca Gobbo, Riccardo Rinnovati, Noemi Romagnoli, Angelo Peli, Chiara Massarenti, Alessandro Spadari, Marco Pietra

**Affiliations:** 1Department of Veterinary Medical Sciences, University of Bologna, Ozzano dell’Emilia, 40064 Bologna, Italy; francesca.gobbo3@unibo.it (F.G.); riccardo.rinnovati2@unibo.it (R.R.); noemi.romagnoli@unibo.it (N.R.); alessandro.spadari@unibo.it (A.S.); marco.pietra@unibo.it (M.P.); 2Department for Life Quality Studies, University of Bologna, Rimini Campus, 47921 Rimini, Italy; angelo.peli@unibo.it; 3Anicura Veterinary Institute of Novara, Granozzo con Monticello, 20060 Novara, Italy; chiara.massarenti@gmail.com

**Keywords:** BAL, cytology, SEA, inflammatory airway disease, horse

## Abstract

**Simple Summary:**

Bronchoalveolar lavage (BAL) fluid cytology is considered the gold standard for the diagnosis of equine asthma both in terms of severity and type of lower airway inflammation. To process BAL fluid, cytocentrifugation is the most frequently employed procedure. The aim of this study was to investigate whether serial BAL cytological samples in horses with severe equine asthma (SEA) under different environmental conditions and medical treatment can undergo significant interpretative differences between two methods of preparation (cytocentrifugation and sediment smear). Considering cytocentrifugation as the reference method for evaluating BAL fluid in cytology, the sediment smear shows poor agreement in the differential cell count for neutrophils as well as lymphocytes and macrophages, with an overestimation of neutrophils and an underestimation of lymphocytes and macrophages. However, our results show that sediment smear, although it seems to be able to recognize the conditions of severe neutrophil respiratory disorders, does not appear overall to be overlapping in terms of differential cell count accuracy.

**Abstract:**

Equine asthma is a common respiratory disease that may affect horses of any age. The diagnosis of severe equine asthma (SEA) (historically referred as recurrent airway obstruction or RAO) is based mainly on the history of the animal and clinical signs, which are further supported by the cytological examination of the bronchoalveolar lavage (BAL). This can also be helpful in monitoring the inflammation of the lower airways in response to environmental management and medication. The cytocentrifugated preparation is usually considered the method of choice for BAL cytological interpretation. The aim of this study was to compare the results in terms of differential cell counts (DCC) in BAL cytology performed on sedimented smears and cytocentrifugated preparations. To carry this out, 48 BAL samples were collected from six horses with SEA that were subjected to a process of exacerbation of the disease by environmental stimuli, which was later followed by the appropriate treatment. Each collected BAL fluid was equally divided into duplicate portions: one processed by cytocentrifugation and one by sediment smear from simple centrifugation. Cytologic examination of all BAL by both methods showed poor concordance in DCC, although it was still able to allow diagnostic recognition of severe lung neutrophilic disorders. These results suggest that sediment smear preparation, although remaining a useful method in general equine practice associated with clinical assessments in the diagnosis of SEA under conditions where there is no possibility of using a cytocentrifuge, cannot be considered a comparable alternative.

## 1. Introduction

Bronchoalveolar lavage (BAL) is currently the most widely employed method to assess the presence and type of cells in the lower respiratory tract. Furthermore, it constitutes a valuable tool both when employed for research purposes and for clinical investigations in horses [1,2].

The widespread use of the BAL owes credit to its considerable versatility in diagnostic lung pathology, often providing a valuable aid in formulating the diagnosis [3]. In fact, it can be employed both for a cytologic assessment of the cell population present in the airways and for analysis of proinflammatory cytokines by biomolecular investigations [4,5,6,7,8,9].

Among the non-infectious inflammatory diseases of the lower airways in horses (recently renamed “Equine Asthma” Syndrome) and common cause of poor performance in the horse, two primary manifestations have been described: a mild/moderate form, also called inflammatory airway disease or IAD; and a severe form, previously referred to as recurrent airway obstruction (RAO) [10,11].

In severe equine asthma (SEA), the analysis of bronchoalveolar lavage fluid represents a fundamental element that, associated with history, clinical examination, and collateral diagnostic investigations (for example, airway endoscopy or lung function evaluation), can confirm and further characterize the diagnostic suspect [10,11,12,13,14]. In horses with SEA, bronchoalveolar lavage fluid cytology is also employed to monitor lower airway inflammation in response to environmental management and medication [15]. 

Moreover, it has been observed that these cases tend to present a moderate to severe neutrophilic lower airway inflammation, which is typically characterized by an increase in BAL neutrophil percentage above 10–25% [16,17,18], and by a concurrent decrease in mononuclear cell percentages (lymphocytes, macrophages) [2,10,12]. However, the proportion of macrophages and lymphocytes has been proven to not be clinically relevant for the diagnosis of SEA [19].

Currently, the “gold-standard” technique for processing BAL and making it suitable for diagnostic evaluation under an optical microscope involves the use of a specific cytocentrifuge for cytological fluid. This machinery still represents a major obstacle for smaller facilities, which therefore have to rely on well-equipped laboratories to send the properly stored samples [1,20].

Only a few studies investigated the use of the smear as an alternative method to cytocentrifugation for BAL processing [1,20]. While cytocentrifugation allows for the uniform concentration of BAL cell compartments in a focal circular area on the slide, sediment smearing involves manually smearing a drop of sediment (cell pellet) after the centrifugation of BAL fluid and subtraction of the supernatant. Therefore, the latter appears unevenly distributed and along the entire length of the slide. Despite the qualitative and quantitative differences between cytocentrifuged and sedimented smear preparations, there are studies demonstrating that the diagnosis of inflammatory lung diseases can be performed by either cytological method [1,20].

We hypothesized that the diagnostic information resulting from the differential cell count of cytological sediment smear could be significantly in agreement with the information provided by the same cytocentrifugated preparations in the presence of severe airway inflammation. Therefore, the aim of this study was to determine, through serial sampling, whether sedimented smear preparations could be used to accurately quantify cellular components present in BAL during inflammatory airway disease of the horse.

## 2. Materials and Methods

### 2.1. Animals and Study Design

All animal procedures were approved by the Ethics and Scientific Committee of the Alma Mater Studiorum University of Bologna in the plenary session on 3 March 2009 (protocol n° 04/55/09) and submitted to the Italian Ministry of Health and conducted in accordance with European legislation regarding the protection of animals used for experimental and other scientific purposes (Council Directive 86/609/EEC). 

This is a prospective study. The same animals were also part of a previous study [21] and the number of samples examined is a convenience sample based on the numbers used in previous studies [1,22].

The trial was performed at the University of Bologna at the Department of Veterinary Medical Sciences and the cytological examination was performed at the pathology laboratory in the same place. The horses that were employed belonged to the Italian Army.

Based on their history, clinical examination and instrumental and laboratory findings [23], six horses affected by SEA for at least one year were enrolled in the study.

All the animals lived outdoors and were in the asymptomatic phase for at least 6 months, during which they had been kept in paddocks and in absence of any pharmacological treatment. 

The choice of the timing of BALs collection was decided in agreement with Ainsworth et al. (2006) and with Tee et al. (2012) [24,25]. So, the BAL were immediately processed and stored, and read at later times.

Upon admission, the horses were hospitalized for 6 days in the stable box, where the doors were kept open with dedusted chip litter to maintain levels of environmental dust at minimum levels. During this period, they were subjected to three bronchoalveolar lavages (BAL): one on the first day of hospitalization, one on the following day and one two days later (T0, T1 and T2) (Figure 1).

Seven days after admission, the horses were placed in a stimulating environment (challenge) by inducing an inflammatory stimulus in susceptible horses. Specifically, the challenging environment involved closing the barn doors, placing a fixed straw bedding that had not been changed and having stable staff turn the bedding once a day to increase environmental dust. The horses were left in this environment for six days. During the challenge, four BAL extractions were performed on all subjects with intervals of 24 h, the first one the day after the start of the challenge (T3, T4, T5 and T6) (Figure 1).

This period was followed by the desensitization phase (lasting 4 days) and the doors of the shelter were opened, the straw bedding was replaced with wood shavings and the subjects were treated pharmacologically with dexamethasone (0.1 mg/kg i.v.) [26]. At the end of the four days, the last sample of BAL was obtained (T7) (Figure 1).

### 2.2. Bronchoalveolar Lavage Collection and Smear Preparation

The patients were sedated to perform bronchoalveolar lavage (acepromazine maleate 2 mg/100 kg; detomidine HCl 1 mg/100 kg) and the procedure was carried out as previously described [27]. Briefly, a Kruuse bronchoalveolar lavage catheter (Kruuse, World veterinary supplier, Denmark; size: 2.5 mm inner diameter, 10 mm outer diameter, length 240 cm) was passed nasally into the distal respiratory tract. A 210 mL aliquot of sterile pre-warmed (37–38 °C) isotonic saline solution, followed by 30 mL of air, was infused and re-aspirated through a 60 mL syringe and then passed into a glass which was kept on ice.

Bronchoalveolar lavage fluid that was recovered from each subject (about 60 mL from each horse) was evaluated macroscopically to ascertain the presence of foam (indication of the presence of surfactant) and to evaluate the turbidity, color and presence of macroscopically visible agglomerates of material. The sample was then immediately refrigerated at a temperature of +4 °C and then centrifuged for smear preparation or cytocentrifugated within 3 h of collection.

For the preparation of the cytocentrifugated samples (Shandon Cytospin 3, Shandon Scientific Ltd., Runcorn, Cheshire, UK) in duplicate, BAL was cytocentrifugated and processed as described in Pickles et al., 2002 [22].

For sedimented smear preparations in duplicate, about 30 mL of BAL was centrifugated for 10 min at 1000 rpm (560× *g*). At the end of the cycle, the supernatant had been removed and the remaining portion of material was mixed gently, and one drop was smeared with a blood smear technique in a slide and left to air dry as described in Pickles et al., 2002 [22].

### 2.3. Cytology Stain and Evaluation Criteria

Slides from both processing methods were stained with May–Grünwald–Giemsa stain (MGG) (cat no. 04-090805, Bio-Optica, Milan, Italy) after drying them in air.

Examination and cytological reading of all slides was performed blindly by a pathologist. The samples were considered suitable for reading if they had a minimum of 50 accounting cells out of the total and an intact and recognizable cellular morphology.

Two types of assessments were made on all preparations:(1)Morphological (cell morphology, homogeneity of cells, and presence of other cells);(2)Cell counts.

The observation in all slides was performed as a first step at small magnification (10×) to evaluate the homogeneity and the presence and amount of mucus. In the evaluation of mucus, both aggregates of free protein amorphous material and neutrophil extracellular traps (NETs) were included together because it was not possible to differentiate the type of protein material with special histochemical staining. The presence of mucus was recorded and semi-quantified, assigning each preparation a score from 0 to 3, where 0 indicates the absence of mucus, 1 indicates mucus presence in up to 30% of the smear, 2 indicates mucus presence from 30 to 60% and 3 indicates mucus presence of more than 60% of the smear. A higher magnification (40×) was used to recognize the morphology and the peculiar characteristics of the different cell types [28]. Cells evaluated in manual differential count were inflammatory cells (neutrophils, mast cells, lymphocytes, macrophages and eosinophils). Moreover, we also recorded the presence and morphological appearance of the epithelial cells, but since they were present in small number and in a very limited quantity of samples, they were excluded from the differential cell count.

From each slide, 5 microscopic fields at 40× were selected and evaluated. The choice of the five fields was based on the homogeneity of the distribution of cells and mucus within the slide. In samples where cells and mucus showed a uniform distribution without the presence of aggregates, the fields were chosen randomly. On the other hand, in samples where there were significant aggregations of cells and mucus (NETs) alternating with parts where these elements were less concentrated, the fields were chosen from either area. 

Each count was averaged and expressed as a percentage of the total cells counted.

### 2.4. Statistical Analysis

Statistical analysis was performed using commercially available software (MedCalc Statistical Software version 12.2.10, MedCalc Software Ltd., Ostend, Belgium https://www.medcalc.org/features/statistics.php, accessed on 1 September 2022). Microsoft Excel 2011 was used for the tables and descriptive statistics. Assessment of the data for normality was calculated by applying the D’Agostino–Pearson test. Data were analyzed using descriptive statistics and expressed as mean ± standard deviation or median and range (minimum–maximum) as appropriate.

The agreement, precision and accuracy of the differential cell count between the cytocentrifugates and the smear evaluation for each variable investigated were assessed using Bland Altman, Pearson’s precision coefficient test and concordance correlation coefficient (ρc) and values of ρc were interpreted according to McBride (2005) [29] (ρc < 0.90: poor; 0.90 to 0.95: moderate; 0.95 to 0.99: substantial; >0.99: almost perfect).

The level of agreement of mucus estimation between cytocentrifugates and sedimented smear was determined using the Cohen’s kappa coefficient (K): poor, k < 0.00; slight, 0.00 ≤ k ≤ 0.20; fair, 0.21 ≤ k ≤ 0.40; moderate, 0.41 ≤ k ≤ 0.60; substantial, 0.61 ≤ k ≤ 0.80; and almost perfect, k > 0.80 [30].

The repeatability of the differential cell count of cytocentrifuged and smear preparations was determined based on the results of the coefficient of variation (CV) test. Briefly, two duplicate pairs of cytocentrifuged and sedimented smear preparations were selected and a differential cell count of 5 areas at 40× for each slide was performed by the principal author 12 times in three different sitting. The CV was then calculated as the ratio of the standard deviation to the mean, expressed as a percentage. The CV for smear preparation was considered acceptable based on traditional values for laboratory tests (<25%) [22].

## 3. Results

### 3.1. Animals and Samples

The animals included in this study consisted of six subjects from mixed breeds and sexes (five Italian Saddle horses and one Appaloosa; 4 females and 2 castrated males) with an average age of 15.7 ± 1.9 years, with body weight ranging from 400 to 520 kg.

A total of 96 slides, 48 cytocentrifuges and 48 smears, were analyzed. These 48 preparations derive from the eight samples obtained from the six subjects (T0–T7). Twelve (12.5%) were considered unsuitable for reading, because either the cytocentrifugate or sedimented smear were poorly cellular (<50 cells as described in the material and methods). Thus, a total of 84 samples were considered suitable and subsequently submitted for cytological reading.

### 3.2. Cell Morphology, Homogeneity of the Samples and Presence of Mucus, Epithelial and Inflammatory Cells

Cell morphology was appreciably better in the cytocentrifugated preparations, where the different cell types were more rapidly and easily recognized. In such preparations, the cells appeared sharper and more defined, the colors were sharper and there was less overlap between different cells. On the contrary, sedimented smear preparations were characterized by cells that were slightly smaller in size, darker in color and with less sharp contours.

Cytocentrifugated preparations were highly cellular and presented as a uniform dot. On the other hand, the smears cells were often distributed in unevenly, alternating portions characterized by a significant presence of cells, to portions of the slide that were almost empty (Figure 2).

The amount of mucus was normally more abundant in the sedimented smear than in the cytocentrifugated smears, with a few single exceptions. In the cytocentrifuged preparations, the mucus usually appeared as aggregates of light pink material against the background of the cellular mat. Conversely, the smear was characterized by conspicuous filaments of a deep pink color that trapped a significant number of cells (Figure 3).

Epithelial cells were observed in three specimens and in minimal amounts (one or two cells at most per slide). Morphologically, they appeared as cells with a high nucleus-to-cytoplasm ratio and rounded nucleus. They showed no atypia. 

Macrophages showed frequent vacuolization, phagocytosed material, and in some cases, they formed multinuclear aggregates (giant cells). The other inflammatory cells present appeared from the normal morphology (Figure 3 and Figure 4).

### 3.3. Mucus Estimation

For each preparation, a semi-quantitative estimate of the amount of mucus present was made (Table 1). The total number of specimens, in which it was possible to estimate the presence of mucus, was 45/48 (94%) for cytocentrifugate and 39/48 (81%) for sedimented smear. In cytocentrifugates, 4/45 (9%) had an amount of mucus of 0, 14/45 (31%) an amount of 1 and 21/45 (47%) an amount of 2 and 6/45 (13%) an amount of 3. On the other hand, in the sedimented smears, 5/39 (13%) presented an amount of mucus of 0, 1/39 (2%) an amount equal to 1, 7/39 (18%) an amount of 2 and 26/39 (67%) an amount of 3. 

### 3.4. Differential Cell Count

Differential cell count (DCC) results for the two methods are summarized as average of counts of each time of sampling in Table 1 and detailed in Appendix A. 

### 3.5. Repeatability of Cell Count

The mean of differential cell count (DCC) and CV value for repeated counts were found to be below the threshold value considered acceptable for the repeatability of laboratory tests (<25%). In fact, in all cases, the CV value was between 0 and 12%, which indicated a very high repeatability of the count (Table 2).

### 3.6. Agreement between Methods

The *concordance correlation coefficient* (*ρc*) calculated for the neutrophil parameter is 0.83. The level of precision based on Pearson’s coefficient and the accuracy considering the bias factor are 0.86 and 0.96, respectively. The Bland Altman plot showed that the DCC of neutrophils tends to be lower in the cytocentrifuge than in the sedimented smear (with a mean difference of −6.7). Moreover, the limits of agreement are high, reaching 22.9 and −36.3, demonstrating that the agreement is poor (Figure 5).

The *concordance correlation coefficient* (*ρc*) calculated for the parameter lymphocytes is 0.61. The level of precision based on Pearson’s coefficient and accuracy considering the factor of bias are 0.71 and 0.87, respectively. The Bland Altman plot showed that the DCC of lymphocytes tends to be higher in the cytocentrifugate than in the sedimented smear (with a mean difference of 6.4). Moreover, the limits of agreement are high reaching 32.2 and −19.4 demonstrating that the agreement is poor (Figure 6). 

The *concordance correlation coefficient* (*ρc*) is 0.59. The level of precision based on Pearson’s Coefficient and the accuracy considering the Bias factor are 0.65 and 0.91, respectively. Bland Altman plot showed that DCC of macrophages in cytocentrifugated and sedimented smear is very close to 0 (with a mean difference of 0.5). Moreover, the limits of agreement are high reaching 19.6 and −18.5 demonstrating that the agreement is poor (Figure 7).

Concordance between the variables “Mast cells” and “Eosinophils”: due to the paucity of data, the concordance between the variables “Mast cells” and “Eosinophils” could not be evaluated. Concordance for the variable “Mucus”: the parameter K used to evaluate the concordance of mucus between the two techniques was found to be 0.21.

## 4. Discussion

In bronchoalveolar lavage fluid samples, the standardization of sampling procedure and cytological interpretation is of crucial importance and is required to compare the results between laboratories. Different studies have described different techniques for BAL collection in equine asthma, in terms of lavage volume, site of sampling, procedure for conservation and preparation of samples [18,19,20,31,32,33,34].

Recently, the use of one single cytology on a pooled BAL fluid sample from both individual lungs showed that this procedure constitutes a valid method in the diagnosis of the inflammatory forms of the lower airway [35].

Cytocentrifugation is the most common method of processing BAL for cytology. The result is a uniform monolayer of cells in one area of the focal length of a common microscope slide [22,35]. However, in veterinary literature, standardized protocols that define speed and time of centrifugation do not exist. In veterinary practice, because cytocentrifuges are quite expensive and require specific equipment, the samples collected from the washing must be properly stored and sent to a private laboratory for processing and preparation. In equine practice, a method of preparing BAL that requires readily available and inexpensive equipment, and that provides acceptable cytological quality (e.g., sediment smears by centrifugation), could be a practical and fair alternative to cytocentrifugation. 

In our study, between the two different types of samples (cytocentrifugated preparations compared to sediment smears) analyzed by light microscopy, we could observe a clear difference in the homogeneity of the cell monolayer. In fact, in the cytocentrifugates, the cells appear more evenly distributed, while in sedimented smears, there are significant differences in the distribution of the cells with more concentrated areas alternating with more rarefied areas. This feature may be partially due to the smeared action during the slide preparation process and to the greater presence of mucus, which tends to trap a certain proportion of cells.

Cell concentration is another element in which there are appreciable differences between the two methods. Cytocentrifuge preparations appear in many cases more cellular at the same viewing magnification than the corresponding smears. This is probably related to the method and the greater concentration capacity of the cytocentrifuge.

Even the cellular morphology was slightly different. In fact, the cells of cytocentrifuge preparations appear much more delineated and recognizable than those of the smears, and of slightly larger in size. This allows the observer a more precise and quicker recognition of the cell category to which they belong. On the other hand, the cells within the smears appear significantly smaller and darker (they retain more dye making the staining more intense) with fewer sharp borders, rendering their interpretation slower.

In the cytocentrifugated preparations, there is also less cellular overlap, which is probably due in part to the intrinsic nature of the method and in part to the frequently lower presence of mucus noted than in the corresponding smears. It is hypothesized that the presence of less mucus is due to a greater loss of the same within the machine during the centrifugation process, also confirmed by the very low K value (0.21) of this variable.

The coefficient of variability (CV) for cell count repeatability presented values between 12 and 0, which is below the acceptable limit based on traditional values for laboratory analysis (<25%), so the repeatability was very good. 

Evaluation of the agreement between the two cytological methods was assessed by multiple tests. The evaluation of the agreement, precision and accuracy between the two methods was carried out with statistical processing involving the calculation of a correlation coefficient (concordance correlation coefficient—*ρc*). This statistical approach is usually employed to assess the degree of concordance between laboratory methods on the same analyte by using the gold standard technique (in our case cytocentrifugate) as a reference. It is an analysis of extreme precision, whose threshold values are very high and set for generic analytical methods. In our case history, in which differential cell counts are compared in the same preparation and numerically expressed as a percentage of the total, values approaching 0.90 are considered as very high. From the results that have emerged, we can state that for granulocyte neutrophils the concordance tends to be high (0.83) and certainly higher than for the other components under evaluation. Precision and accuracy (0.86 and 0.96, respectively) were observed to be very high. Lymphocytes (*ρc* = 0.61; precision 0.71; accuracy 0.87) and macrophages (*ρc* = 0.59; precision 0.65; accuracy 0.91) have less concordant values. These data partially agree with what has been reported in the literature [1,20]. However, our results may be only partially overlapping because of the different statistical procedures chosen to compare the methods, the different cell count methods applied and the number of samples analyzed. When comparing the two methods of BAL processing, carried out in subjects with IAD, Lapointe et al. [1] found an increase in the percentage of neutrophils while the percentage of lymphocytes appeared significantly lower on cytocentrifuged specimens. Pikles et al. (2002) [22] evaluated 13 samples from five subjects with SEA, in which neutrophilia was comparable between smear and cytocentrifugated preparations. By contrast, they found significant differences in macrophage and lymphocyte counts, although these were considered not relevant for the diagnosis. 

In our case, Bland Altman’s graph showed poor agreement of DCC results for all categories of cells evaluated. The limits of agreement were found to be high in neutrophils (22.9 and −36.3) and lymphocytes (32.2 and −19.4), and tend to be lower for macrophages (19.6 and −18.5). Furthermore, the mean difference of macrophages was close to 0; neutrophils revealed a mean difference of −6.7, showing overestimation in the smear. By contrast, an underestimation was observed for lymphocytes in the smear (mean difference +6.4).

These differences are mainly due to intrinsic factors in the smear method, which has a non-uniform distribution and also depends on the amount of mucus in the sample.

Regarding the other categories of cells, the presence of mast cells and eosinophils was found to be low. Although present in low numbers in all samples evaluated, mast cells were easily identified by their characteristic granules highlighted by the May–Grünwald–Giemsa stain used. This did not allow the data to be used for statistical analysis for comparison. The low number of counted mast cells can be partially attributed to a greater ease of damage of these cells in the smears. For greater reliability of the results, McGorum et al. (2007) [14] suggested counting a number of cells equal to at least 300 cells in each field which is not possible at high magnification [14]. By contrast, a recent study comparing two types of differential mast cell counts suggested that for this cellular compartment, the best method of cytocentrifugate assessment is evaluation in 5-field method on slide at 40× [19]. Based on this, we decided to use high microscope magnification (40×) in order to identify cell morphology with accuracy and precision.

Epithelial cells in this study were excluded from differential cell counts. To date, no consensus has been established on epithelial cell inclusion during the cytology reporting of BAL sampling. A recent article on the possible effects of epithelial cell inclusion or exclusion in BAL tracheal lavage cytology indicated that the presence of epithelial cells can influence differential cell counts and suggested the importance of making whether they are included or excluded from the count explicit [34]. 

The data show poor agreement in differential cell counts and percentage values collected for neutrophils, lymphocytes and macrophages, and thus the two methods are not comparable in precision and accuracy in differential cell counts. Therefore, our results suggest cautious use of the sediment smear in BAL and express a limitation in the possibility of using it as an alternative, especially in studies and experimental trials on respiratory diseases in the horse.

Moreover, by sedimented smear, it was possible to find a neutrophil percentage value (>15%) that if paired with the history, clinical examination and possibly endoscopy helps to diagnose a patient as suffering from severe neutrophilic lower airway disease.

The results for the repeatability test of the counts calculated by CV validate internal validity, but external validation cannot be assessed, and this represents a limitation.

Unfortunately, these preliminary results are not currently supported by lung function data. Thus, it is not possible to compare the clinical stages of the disease exacerbation and remission and the effect of therapy in the BALs analyzed in this study. Further studies are required to demonstrate variations in BAL cytology in patients at different phases of SEA. 

## 5. Conclusions

In conclusion, we can state that for more in-depth studies that require greater precision in identifying the exact number of cells contained in the preparation, cytocentrifugation remains the reference standard technique. In fact, cytocentrifugated BAL appears morphologically more efficient, easier and quicker to read, although less representative of the amount of mucus, and still remains the method of choice for studies and research on respiratory diseases in the horse.

In the face of high equipment costs, the smears do not show differences that would invalidate the recognition of severe neutrophilic airway inflammatory diseases. Therefore, it could be a practical and less expensive alternative to cytocentrifuge preparations in general equine practice where there is no possibility of using a cytocentrifuge, at least for the diagnosis of severe neutrophilic airway inflammatory diseases. 

Further studies including both healthy control cases and horses with different types of lung disorders are necessary to confirm our results and to expand the studies to other respiratory diseases in the horse.

## Figures and Tables

**Figure 1 vetsci-10-00527-f001:**
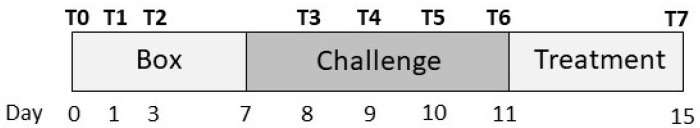
Timeline flow chart of trial.

**Figure 2 vetsci-10-00527-f002:**
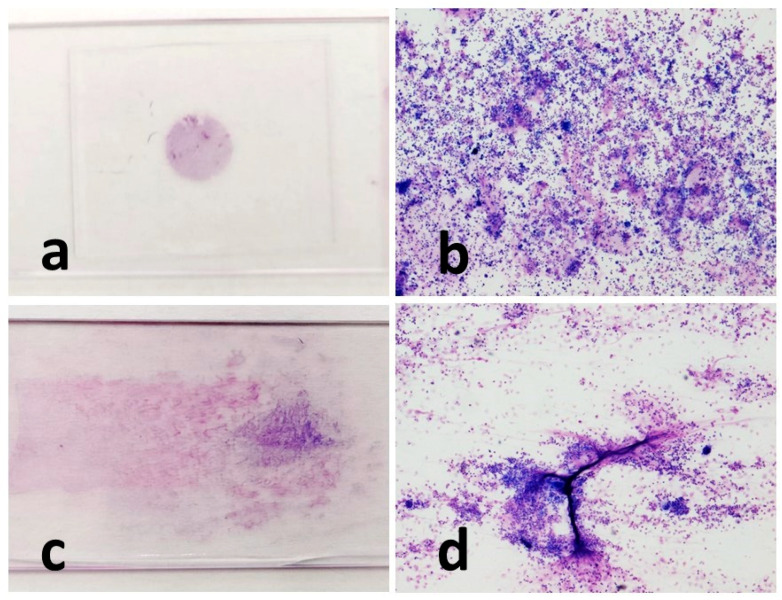
Macroscopic and microscopic appearance of the cytocentrifugated (**a**,**b**) and sedimented smear (**c**,**d**) preparations of the same horse (T6 treatment). Macroscopically, the cellularity of the cytocentrifugate appears concentrated in a small circular area (**a**), and microscopically, the cells appear homogeneously distributed (**b**). The cellularity of the sedimented smear instead has seeped through the glass (**c**). Microscopically, cell aggregates of different densities can be observed. The cells appear unevenly distributed with the presence of mucus (in bright pink) which retains a large number of cells (**d**). MGG stain, 4× magnification (**b**,**d**).

**Figure 3 vetsci-10-00527-f003:**
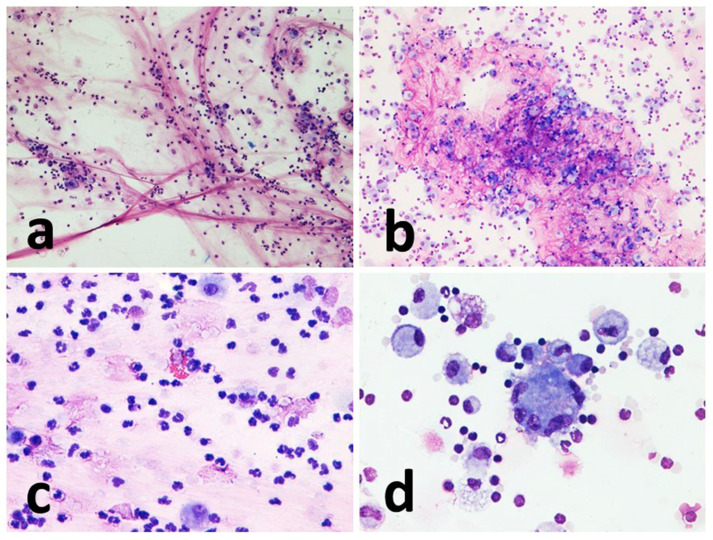
Representative images of sedimented smear and cytocentrifugated preparation of the same horse at different time points of the study. (**a**) Sedimented smear (T6). Numerous intertwined strands of mucus (deep pink in color) are seen, entangling isolated inflammatory cells (predominantly neutrophils). (**b**) Cytocentrifugated preparation (T5). Voluminous aggregate of faintly pink amorphous material (mucus) mixed with cells mainly represented by lymphocytes and macrophages. (**c**) Sedimented smear (T4). A heterogeneous population consisting mainly of lymphocytes, neutrophils and macrophages and thin filaments of mucus. An eosinophil is found in the center of the figure. (**d**) Cytocentrifugated preparation (T2). Uniformly distributed macrophages, lymphocytes and neutrophils are observed, a multinucleated giant cell is found in the center. MGG stain, magnification 10× (**a**,**b**), and 40× (**c**,**d**).

**Figure 4 vetsci-10-00527-f004:**
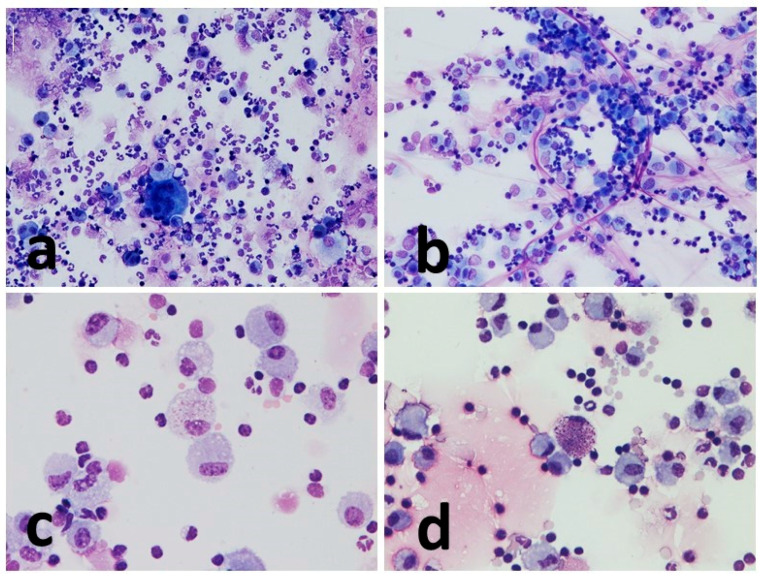
Representative images of sedimented smear and cytocentrifugate of the same horse. (**a**) Cytocentrifuged preparation (T6). Mixed inflammatory cells with a prevalence of neutrophils. A multinucleated cell can be seen in the center. (**b**) Sedimented smear (T6). Inflammatory cellular components are mainly represented by lymphocytes and macrophages. Thin filaments of mucus are seen in pink. (**c**) Cytocentrifuged preparation (T2). Many foamy macrophages are observed. A macrophage that appears engulfed by pink granular material (proteinaceous) is visible in the center. (**d**) Microscopic appearance of T2 cytocentrifuge. A mast cell is seen in the center of the picture. MGG stain, magnification 20× (**a**,**b**) and 40× (**c**,**d**).

**Figure 5 vetsci-10-00527-f005:**
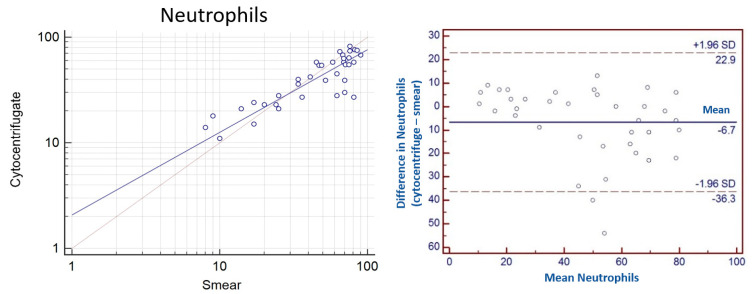
(**Left**): Scatter plot of the mean of neutrophils of cytocentrifugated DCC and sedimented smear DCC (expressed in logarithm) showed a moderate positive correlation. (**Right**): Bland Altman plot of agreement between difference and mean in neutrophils from cytocentrifugated and smear preparations of equine BALF. Horizontal lines indicating limits of agreement.

**Figure 6 vetsci-10-00527-f006:**
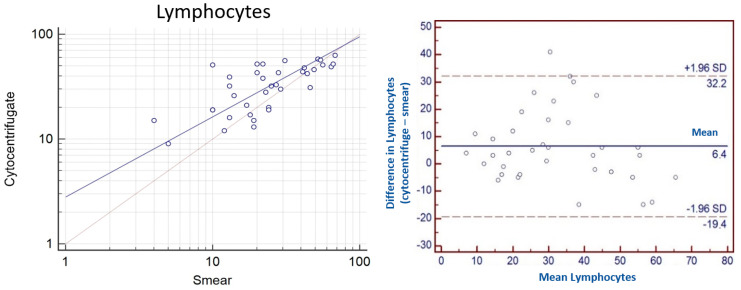
(**Left**): Scatter plot of the mean of lymphocytes for cytocentrifugated DCC and sedimented smear DCC (expressed in logarithm) showed a slight positive correlation. (**Right**): Bland Altman plot of agreement between difference and mean in lymphocytes from cytocentrifugated and smear preparations of equine BALF. Horizontal lines indicating limits of agreement.

**Figure 7 vetsci-10-00527-f007:**
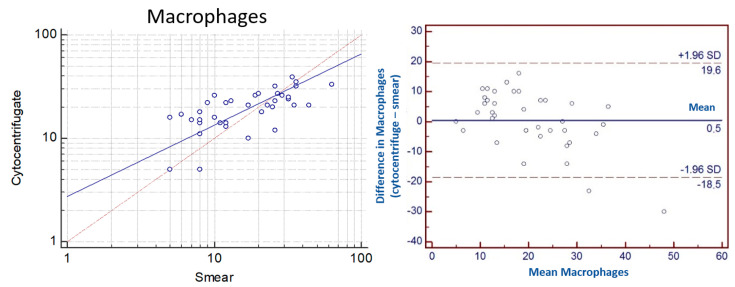
(**Left**): Scatter plot of the mean of macrophages of cytocentrifugated DCC and sedimented smear DCC (expressed in logarithm) showed a slight positive correlation. (**Right**): Bland Altman plot of agreement between difference and mean in macrophages from cytocentrifugated and smear preparations of equine BALF. Horizontal lines indicating limits of agreement.

**Table 1 vetsci-10-00527-t001:** Summary of the results for the DCCs of neutrophils, lymphocytes, macrophages and mucus evaluation. Abbreviations: CC, cytocentrifugate; SS, sedimented smear.

Time		T0	T1	T2	T3	T4	T5	T6	T7
**Neutrophils** % Mean ±(SD)	CC	37.4 ±(26.4)	44.75 ±(19.7)	25.5 ±(17.2)	54.7 ±(19.6)	50.0 ±(22.6)	41.3 ±(24.7)	44.2 ±(21.2)	49.7 ±(18.7)
	SS	40.0 ±(26.4)	60.8 ±(24.1)	15.5 ±(8.3)	71.0 ±(18.9)	51.4 ±(33.1)	49.2 ±(24.6)	55.8 ±(23.2)	44.0 ±(16.5)
**Lymphocytes** % Mean ±(SD)	CC	38.0 ±(15.0)	37.5 ±(18.0)	49.3 ±(15.7)	28.3 ±(16.8)	30.7 ±(14.8)	36.2 ±(17.0)	33.3 ±(15.1)	31.3 ±(11.8)
	SS	29.3 ±(18.6)	23.0 ±(17.3)	60.0 ±(7.3)	17.5 ±(12.9)	26.4 ±(22.5)	29.8 ±(15.8)	26.2 ±(15.0)	29.0 ±(9.9)
**Macrophages** % Mean ±(SD)	CC	28.3 ±(7.8)	16.3 ±(2.5)	24.3 ±(7.8)	11.3 ±(3.3)	22.2 ±(10.1)	20.8 ±(11.2)	18.0 ±(8.2)	26.8 ±(7.9)
	SS	28.3 ±(7.3)	16.3 ±(18.5)	24.3 ±(9.9)	11.3 ±(7.0)	22.2 ±(23.3)	20.8 ±(10.9)	18.0 ±(11.6)	26.8 ±(7.1)
**Mucus** Median (min–max)	CC	1 (0–2)	1 (1–2)	2 (0–3)	2 (1–3)	1 (1–2)	3 (1–3)	2 (1–2)	2 (2–3)
	SS	2 (0–3)	3 (3–3)	0 (0–3)	2.5 (0–3)	3 (2–3)	3 (3–3)	3 (2–3)	2 (2–3)

**Table 2 vetsci-10-00527-t002:** Cell count repeatability test. For each cell category of the four samples, the mean, standard deviation (SD) and coefficient of variation (CV) are expressed as a percentage.

I	Cytocentrifugate	Smear
Mean neutrophils (%)	58.7 (±2.7)	57.7 (±2.4)
Neutrophils CV (%)	2	1
Mean lymphocytes (%)	15 (±0.3)	13.3 (±0.4)
Lymphocytes CV (%)	12	4
Mean macrophages (%)	26.3 (±0.8)	29 (±1.6)
Macrophages CV (%)	0	0
**II**	**Cytocentrifugate**	**Smear**
Mean neutrophils (%)	24 (±1.6)	16.3 (±0.9)
Neutrophils CV (%)	0	7
Mean lymphocytes (%)	42 (±2.1)	19.3 (±1.8)
Lymphocytes CV (%)	4	6
Mean macrophages (%)	34 (±2.3)	64 (±3.3)
Macrophages CV (%)	5	3

## Data Availability

All the data generated or analyzed during this study are included in this published article. The raw datasets used and analyzed during the current study are available from the corresponding author on reasonable request.

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
