# Peer review of "Bronchoalveolar Lavage Cytology in Severe Equine Asthma: Cytocentrifugated versus Sediment Smear Preparations†"

_vetsci, 2023, doi:10.3390/vetsci10080527_

Round 1

Reviewer 1 Report

Please see the attached document

Extensive editing of the English language is required. Even though the meaning of most sentences can be understood, the wording used is often inappropriate for what is considered standard "scientific English." Several typos and grammatical errors are encountered throughout the manuscript and require careful revision by the authors. Consulting with an English native speaker is recommended to facilitate editing. 

Author Response

We have uploaded two versions of the revised manuscript. One with traceable changes, and a clean version for easier reading.

Reviewer 2 Report

Dear Authors, please correct the marked sentences!

Author Response

Thank you very much for your suggestions. The manuscript have been changed following also your indications.

We have uploaded two versions of the revised manuscript. One with traceable changes, and a clean version for easier reading.

Reviewer 3 Report

I appreciate the big work behind the paper and I think data obtained are precious. Nevertheless, there are several issues to address. In fact the methods are not clearly explained; the discussion is fairly supported by results; the literature review is incomplete. Please find some specific comments;

Line 60: This is not true. BAL is not the sample of choice for microbiological examination, as the TW is. Performing a BAL in case of an infectious disease such as pneumonia can be dangerous for the horse or non diagnostic. Please eliminate this sentence.

Line 76 - 80: Please rephrase and report the values of the differential count of inflammatory cells. Please refer to the two consensus statement of Inflammatory Airway Disease and Equine Asthma (Couetil et al 2007, Couetil et al 2016), that cannot be ignored in this introduction.

Line 87-92: Please cite the studies just once. I am not sure that reference 21 is the correct one (perhaps  33?). Moreover reference 21 and 34 are the same, and the year of reference 33 is 2002, not 2010.

Line 96 -107: I think all this part should be removed, as it is a preview of the conclusions of the study.

Line 110: the acronym EAs is referred to Equine Asthma Syndrome again as EAS? As these horses are affected by a severe form of Equine Athma, I suggest to change it to sEA (Severe Equine Asthma), and I think it should be more appropriate also in the introduction section.

Line 126: This is speculative, because you do not have any measurement of lung function.

Line 152-154: I do not understand how the cytocentrifugation was performed. With 15-20 ml? Have you a reference for that?

Line 155 - 158: Same for smear. Was the surnatant discharged and the pellet resuspended? Please clarify this point and add a reference if possible.

Line 166-177: The method used for the differential count of the inflammatory cell has not been described or cited.

Line 199-200: Please do not repeat the data of population.

Line 204-205: you already have explained this feature

Line 206: A lot of samples were not diagnostic. Why? It was more for cytocentrifuged or smear? I think it is an important issue.

Line 259-262: The literature regarding BAL of horses suggests to count at least 300 cells for differential count. Moreover, I think the tables in supplementary are unnecessary. I suggest to report in a table the median values of the two groups (eliminayting the graphs) or to report the results in the text and conserving the graphs.

Line 270-272: This part should be moved to statistical analysis section

Line 301-304: In the material and methods section, this kind of analysis is not reported

Line 305-310: Same of the previous paragraph.

Line 322-324: This sentence is unnecessary for the aim of the study

Line 325-336: Same of the previous paragraph. The difference between sample sites is absolutely irrelevant in this paper

Line 398 - 405: As authors state, the differential count of the inflammatory cells of the BAL, that is the most important feature for the diagnosis of Equine Asthma, has not been performed in the most accurate way. Moreover, the absence of mast-cells and eosinophils represent a limit of this study. Therefore, I do not agre to emphasize the results obtained suggesting the reader that there is no difference between cytocentrifugation and smear, because this is not the result of the study. You can just say that the neutrophils count have a good correlation. You cannot suggest to substitute the cytocentrifugation with the smear.

412-420: As above

421 - 430: This paragraph is not understandable. Please rephrase or eliminate (suggested)

431 - 439: I do not agree that little is known. Several paper from Dr. Lavoie of Montreal University have focused on it. Please update the literature and rephrase.

440-451: Please add in the results session a table reporting this data (see comment above).

455-464: this is a too ambitious conclusion

Please revise the English form of the paper. There are no grammar error, but several sentences are cumbersome, too long and too difficult to understand.

Author Response

We have uploaded two versions of the revised manuscript. One with traceable changes, and a clean version for easier reading.

Please see the attachment for the point by point response

Reviewer 4 Report

This report addresses Bronchoalveolar Lavage Cytology in Severe Equine Asthma Syndrome during different stages of the disease and also with two different methods for the cytology evaluation. The topic is interesting in equine medicine as equine asthma is a common disease in these patients.

The length and structure are correct, and in general terms, it is effortless to understand. However, there is some aspects that should be clarify before publishing it.

Introduction

- Line 59: respiratory tree. With the BAL we are looking for the alveolar cells, not cells from the respiratory tree. Moreover, as you describe in the material and methods, when you perform a BAL the way to know if is from the alveolar area is the presence of foam (surfactant) from the lungs not from the airways.

- Line 60. Usually, the samples from cytology are not used for culture because there is contamination from the upper airways when you pass the BAL catheter. When you want to get samples for culture, is preferable perform a transtracheal wash.

- Line 73-75. Currently, little is known about the degree and type of BAL fluid inflammation, during the different phases of the disease (asymptomatic, exacerbation and remission phase). There are some articles describing not differences between the cytology of BALF with different treatments and stages of the diseases.

Davis KU, Sheats MK. Bronchoalveolar Lavage Cytology Characteristics and Seasonal Changes in a Herd of Pastured Teaching Horses. Front Vet Sci. 2019 Mar 14;6:74.

Léguillette R, Tohver T, Bond SL, Nicol JA, McDonald KJ. Effect of Dexamethasone and Fluticasone on Airway Hyperresponsiveness in Horses With Inflammatory Airway Disease J Vet Intern Med. 2017 Jul;31(4):1193-1201. They said that there are no significant effects on the clinical signs or the number of inflammatory cells (except lymphocytes) in BALF.

- Line 92. The reference 21 conclude that you can use smear as diagnosis tool, so I think they don’t invalidate the technique for diagnosis. This sentence should be rewritten.

Materials and Methods

- Line 109. Eleven animals were used in the study. Seems that there a mistake in the number of the horses because in the rest of the text there are only 6 animals.

- Line 113. How and when was performed the diagnosis of asthma?

- Cytology stain and evaluation criteria. Who perform the evaluation of the cytological preparations? Was a pathologist?  Was just one person or more people? Was the study blinded?

Please, include this information in the section.

Results

- Line 306. Was the repeatability done with the same operator? I meant, it is important to know if both samples were evaluated for the same person.

Discussion

- Line 426-427. …the inflammatory pattern should completely normalize during periods of remission… Some of inflammatory markers as interleukins can be modified during the different stages of the disease but not the neutrophil counts. That is the reason why BAL is the gold standard for the diagnoses of asthma, but it cannot be used as monitorization of the treatment. Please rewrite the sentence to clarify this problem.

- Line 427-430. There is a mistake, the sentence is repeated.

However, if on the one hand the BAL fluid cytology is able to disclose neutrophilic inflammation during the exacerbation of the disease [18], on the other hand, the inflammatory pattern should completely normalize during periods of remission, induced by antigen avoidance strategies [36,37].

Could be great to include a couple of sentences about the limitations of the study in this section.

Author Response

(The authors gave the same response as above.)

Round 2

Reviewer 1 Report

I appreciate the work that the authors have done so far on the quality of the English language, which is significantly improved compared to the first version. However, several sentences still have very unusual structures and include unclear wording. It is very hard for the reader to understand the meaning of certain sentences. Overall, more English editing is required.

Author Response

Thank you very much for your effort in this revision. It help us a lot and we think our paper has now been improved.

Reviewer 3 Report

The correction performed by the authors improved significantly the value of the paper. Nevertheless, there are some important issue that should be addressed. Please find specific comments below:

Simple Summary:

Line 15 "still  considered the most efficient method of" change with: "is considered the gold standard for the diagnosis".

Line 21:  Eliminate "of the collected BAL.

Line 22: Eliminate "for all the cytological compartment"

Line 23: Eliminate "these two methods are equally comparable in terms of results." Rephrase as: These findings show that, although somewhat slightly more probing in terms of reading, ...."

Line 26: Eliminate "commercial"

Abstract:

Line 28: Eliminate "form"

Line 32: "In order to investigate whether in the BAL cytology in horses with SEA under different environmental conditions and before and after treatment there are interpretative differences between two methods of smear preparation of the collected fluid" Rephrase as " In order to investigate the interpretative differences between two method of smear preparation of BAL fluid collected from horses with SEA under different environmental and therapeutic conditions, a study"

Line 36: Eliminate "in 8 withdrawal"

Line 36: Eliminate "process exacerbation through"

Line 40: "appears"

Line 40: Eliminate "equally diagnostically"

Introduction

Line 4: Add "respiratory tract of horses"

Line 49-50: Please eliminate this sentence because is unnecessary for the aim of the study.

Line 57: Please move "have been described" after "two clinical manifestations".

Line 61: Please report the meaning of the acronym SEA before using it.

Line 66 - 68: I do not agree with this sentence. Please refer to Bullone et al, 2017 (Scientific Reports), and Leclere et al 2011, Am J Respir Cell Mol Biol, or eliminate this sentence.

Line 75: Remove "," after of

Line 79-84: Rephrase as: "About equine BAL fluid sampling and preparation, literature reports only very few studies exploring an alternative to the use of cytocentrifuge [1,21]. However in these studies, despite the qualitative and quantitative differences between cytocentrifugated and sedimented smear preparations, the diagnosis of inflammatory lung disease was not invalidated [21].

Line 86: Eliminate "on a large cohort of samples set up under the same conditions from" and add "of"

Line 90: Eliminate "(less expensive and more accessible to practitioners) " because is a discussion topic. Eliminate" in clinical practice"

Material and Methods:

Line 95: Please rephrase as: "Six horses affected by SEA for at least one year on the basis of history, clinical examination and instrumental and laboratory  findings (22) were enrolled in the study.

Line 97:  Eliminate: "had a history of signs of recurrent acute onset of asthma "

Line 140: Why 30 ml? The paper of Pickle et al 2002 thtat you are citing reports the use of 100 microL. Please check and explain, this is a very important issue

Line 143: Do not use rpm. Please report g value.

Line 152-153: please remove the sentence, is a repetition of the previous one.

Line 159: NETs is an acronym that has never been used before, please explain.

Results:

Line 213: These 12 slides that were not readable are from cytocentrifugation or sedimentation? Please report the data (and please discuss accordingly)

Line 216: 96-12=84. So why 82????

Line 255: Remove the percentage

Line 315: The percentage of mucus grade 2 is missing

Discussion:

Line 392-396: I think this should be a unique sentence.

Line 404-407: This sentence is not clear. Please rephrase.

Line 409: Less expensive. I think ", which" should be eliminated

Line 407-417: This paragraph should be removed, as this is a personal observation and it is not supported by results. moreover, it is far from the aim of the study.

Conclusion:

Line 439-441: This sentence should be removed.

English language has improved, I corrected some sentences that were cumbersome.

Author Response

Please see the attachment (PDF)

Round 3

Reviewer 1 Report

GENERAL COMMENT

Thanks to the authors for the rebuttal letter and revision. I still think there are a couple of flaws in the study that need to be addressed prior to recommending acceptance:

-          In this version of the manuscript, the authors implemented a new statistical measure of agreement (the ICC) that is not appropriate for the study performed (see my comments below) and adopted it as the main indicator of agreement. Overall, the CCC and the Bland Altman are more appropriate for the type of study performed and should be used to assess the agreement, while the ICC should be removed. Also, there is still confusion in the description and interpretation of the Bland Altman plots.

-          The authors still tend to “overstate” the interpretation of their results, and they do not underline sufficiently the limitations of their study, which are significant. While I believe that the study contains pieces of information that are worth to be published, I also think that objective interpretation of the results is as important as the results themselves for manuscript acceptance.

SPECIFIC COMMENTS

Line 21-22: based on the overall results of the study, this is an overstatement, and it is misleading for the readers, so it should be reworded (see in the comments below why).

Line 50: “No printed form” is a typo?

Line 78-79: Rewording suggested: “Only a few studies investigated the use of methods alternative to cytocentrifugation for BAL processing.”

Line 79-82: It might be helpful to follow up with a very brief description of the technical procedure and methodological differences between cytocentrifugation and sedimentation.

Line 85: Implement the sentence with “through serial sampling” to highlight that multiple samples were taken over time.

Line 96-99: see the comment about power analysis in my previous review, which has not been addressed thoroughly. The authors' answer in the rebuttal letter needs to be implemented in the text (i.e., that is a convenience sample based on numbers used in previous studies, with reference). Again, please state the type of study (prospective/retrospective? Experimental/observational?), the period in which the study was performed (year or years), the facilities where both the clinical and the laboratory phase of the study was performed, and the origin of the horses (research/teaching herd?).

Line 132: what is the meaning of “ca”?

Line 136: centrifuged? Should it be “centrifuged or cytocentrifuged”?

Line 140: was “centrifuged” instead of centrifuge?

Line 175-177: this can be removed from here since it is repeated later in the statistical analysis.

Line 186-194: I appreciate the author's effort in revising the statistics section and the reply with explanations. However, the rationale underlying the statistical analysis still remains confusing to me. In particular, the choice of implementing the ICC is controversial. The study aims to measure the agreement between two continuous variables (the cell counts), and only one pathologist is making the counts. Since the interest is in comparing the agreement of two measurement methods (technique 1 vs. technique 2) applied on the same samples by the same pathologist, the CCC is the appropriate statistical measure. The author is not trying to account for differences between raters (which might warrant an Intraclass Correlation Coefficient or ICC). I do not see the value of evaluating both for this particular study, and, in my opinion, if one measure needs to be chosen, the CCC is the more appropriate one.

Line 188: avoid using acronyms at the beginning of a sentence (ICC)

Line 203: the sentence has unclear wording. The CV is the ratio of the standard deviation to the mean, expressed as a percentage.

Line 204: Avoid using acronyms at the beginning of a sentence.

Line 276: What is the mean DCC? (this is the first time this acronym is used, write it as extended or introduce it earlier in the text)

Line 277-278: this sentence is confusing because it is likely missing the correct unit of measure (%). Was the CV between 0 and 1 % or between 0 and 1? Because the CV is a ratio, one could express it as a decimal or a %. That said, if the CV is expressed as % and between 0% and 1%, it is really good. Instead, if the CV is expressed as decimal and between 0 and 1, it just means it varies between 0% and 100%, which is not good.

Table 1: Implement the table with values of standard deviations (it is reported as one of the parameters in the table heading but not actually included in the table).

Line 285-286: I would recommend placing descriptive statistics before the repeatability assessment, i.e., keep three separate results sections: a) descriptive statistics, b) repeatability assessment, c) agreement between methods

Line 291-292, line 304-305, line 317-318: consider in relation to the comment provided above for the section of statistical analysis (i.e., ICC is not the most appropriate measure of agreement for this type of study)

Line 295-298: the description of the plot is unclear, and some wording is not correct. The followings are my comments regarding the interpretation of the plots:

a)     Neutrophil count tends to be lower on cytocentrifuged samples than sedimented on average. On line 295, I would replace the statement “is lower” that the author uses with a “tends to be lower”. This is because there are still several data points for which the difference is above zero (i.e., cytocentrifuge count is slightly higher than sedimented). It seems that the neutrophil counts tend to be lower on cytocentrifuged, especially for counts that are higher than 40%

b)     The plot data almost have a funnel shape, with the more prominent spread of the data for high mean counts. In line with what was mentioned above, for higher counts (>40%), there is less agreement between the two methods (i.e., there is a bigger difference).

c)     Line 296-298: this sentence is confused and confusing. If the author refers to the two points that are below the low horizontal line (-1.96 SD), then the statement is wrong. The two horizontal lines on the plot are not the confidence intervals but the limits of agreements, calculated as the mean difference ± 1.96 times the standard deviation of the differences. These limits indicate where one expects 95% of the differences between the two methods to lie. The limits of the agreement provide a range in which most of the differences between the two methods fall. The two points that the authors mention are OUTSIDE (not WITHIN) the LIMITS OF AGREEMENT (not the CONFIDENCE INTERVALS) and can be considered outliers.

d)     The narrower the limits of agreement are, the better the agreement is. Does the author think the limits of agreement for the neutrophil count are clinically acceptable? (i.e., does the author think that it is clinically acceptable that the sedimented technique might overestimate the cytocentrifuged technique up to 36.3% and underestimate it up to 22.9%?). This is a very important piece of information for the plot interpretation, and the author should provide it.

Line 308-311 and 320-324: as mentioned for the neutrophil count, the description is confused and confusing. I recommend rewording as suggested above and providing interpretation along with what was pointed out in the comment for the neutrophil count. In particular, does the author think that the limits of agreement are narrow enough to be clinically acceptable?

Line 358: “Creeping” is not the correct word in this context.

Line 362: this observation does not come from statistical analysis but from subjective evaluation; thus, the word “significantly” should be removed.

Line 377-379: as previously suggested, reword the sentence; it is unclear. One cannot use the coefficient of variation as calculated by the author for a comparison of the two methods.

Line 381-384: This should be removed. In my opinion, the ICC is not the appropriate statistical measure for this type of study.

Line 403-406: implement interpretation of the Bland Altman with the clinical relevance of the limits of agreement. The number of outliers is not a relevant criterium to determine whether there is enough agreement or not. The most important criteria are 1) the bias (mean difference) and 2) the width of the limits of agreement. Are the limits of agreement narrow enough to not impact clinical evaluation? (… in my opinion, they are not, and the fact that the sedimented technique might overestimate by up to 36.3% or underestimate up to 22.9% the neutrophil % could be clinically relevant, considering the thresholds that are currently accepted for defining equine asthma).

Line 425: typo “no printed form”

Line 427-428: this sentence is extremely non-scientific and needs to be removed. If reproducibility was not evaluated, the author cannot and should not believe in anything. Also, as suggested already in the previous review, implement this statement by appropriately addressing the lack of reproducibility as a limitation (i.e., the study might have internal validity, but the external validity and overall clinical utility cannot be evaluated)

Line 440: typo “no printed form”

Line 436-437: this sentence is an overstatement based on the results of the study and needs to be reworded.

Line 451-453: I think this is an overstatement. The sedimented technique is an alternative but has important limitations that need to be considered if elected for BAL cytological assessment. 

The English quality has been greatly improved, and the manuscript flows well. However, some sentences have unusual vocabulary or wording, and there are still a few grammatical errors and typos. The author can work on those details with the editor.  

Author Response

For the answers please see the attachment.

Changes and added parts have been highlighted in the text in red font to make reading clearer and faster without generating confusion with previous revisions.

Line references in the text refer to the PDF version.

Reviewer 3 Report

Thank you for the effort in reviewing the manuscript. I think the paper now is suitable for publication.

Author Response

Thank you so much for reviewing our paper.